# Cationic Pillar[6]arene Induces Cell Apoptosis by Inhibiting Protein Tyrosine Phosphorylation Via Host–Guest Recognition

**DOI:** 10.3390/ijms21144979

**Published:** 2020-07-15

**Authors:** Can-Peng Li, Yu-Xun Lu, Cheng-Ting Zi, Yu-Ting Zhao, Hui Zhao, Ya-Ping Zhang

**Affiliations:** 1Key Laboratory of Medicinal Chemistry for Natural Resource, Ministry of Education, School of Chemical Science and Technology, Yunnan University, Kunming 650091, China; lcppp1974@sina.com (C.-P.L.); zyt94817@163.com (Y.-T.Z.); 2State Key Laboratory for Conservation and Utilization of Bio-resource, School of Life Science, Yunnan University, Kunming 650091, China; l273581722@163.com; 3State Key Laboratory of Pu-er Tea Science, Ministry of Education in Yunnan, College of State Science, Yunnan Agricultural University, Kunming 650201, China; chengtingzi@163.com; 4Key Laboratory of Genetic Resources and Evolution, Kunming Institute of Zoology, Chinese Academy of Sciences, Kunming 650223, China

**Keywords:** pillar[6]arene, protein tyrosine phosphorylation, molecular recognition, apoptosis

## Abstract

We reported for the first time that cationic pillar[6]arene (cPA6) could tightly bind to peptide polymer (MW~20–50 kDa), an artificial substrate for tyrosine (Tyr) phosphorylation, and efficiently inhibit Tyr protein phosphorylation through host–guest recognition. We synthesized a nanocomposite of black phosphorus nanosheets loaded with cPA6 (BPNS@cPA6) to explore the effect of cPA6 on cells. BPNS@cPA6 was able to enter HepG2 cells, induced apoptosis, and inhibited cell proliferation by reducing the level of Tyr phosphorylation. Furthermore, BPNS@cPA6 showed a stronger ability of inhibiting cell proliferation in tumor cells than in normal cells. Our results revealed the supramolecular modulation of enzymatic Tyr phosphorylation by the host–guest recognition of cPA6.

## 1. Introduction

Molecular recognition plays a crucial role in biological systems. Although several recognition principles are well understood, the design and synthesis of molecules to recognize expected target molecules, such as proteins and enzymes, continue to pose challenges in the fields of biochemical pharmaceuticals and medical diagnostics. In particular, the biomolecular interactions of peptides and proteins with macrocyclic receptors are of great interest in chemical biology and therapeutic applications [1,2,3,4,5,6,7,8,9,10,11,12]. Recent studies have revealed the host–guest recognition between peptides and macrocyclic hosts, such as cucurbituril, cyclodextrin, and calixarene, on the basis of amino acid sequence-specific recognition via supramolecular interactions, including hydrogen, charge transfer, and stacking interactions [13,14,15,16,17,18,19,20,21].

Pillararenes, a new class of macrocyclic hosts, have recently received considerable attention for their intriguing and unique characteristics and properties, including facile synthesis, easy and simple functionalization, and ready solubility in aqueous solutions or organic solvents [22,23,24]. Although the molecular recognition of pillararenes and guests in organic solvents has been extensively investigated, few studies have focused on the interactions of pillararenes and biomacromolecules (e.g., proteins or enzymes) in aqueous solutions [25,26,27,28,29]. Yang’s group recently found that carboxylated pillar[6]arene can remarkably inhibit the pentamer formation of human papillomavirus (HPV) L1 via selective binding to basic amino acids at the capsid protein interface [29]. This interesting result has laid the foundation for the development of new supramolecular inhibitors for HPV treatment. More recently, Barbera and co-workers reported an antimicrobial coating capable of the sustained release of antibiotics based on electrostatically assembled multilayers between carboxylated pillar[6]arene and poly(allylamine hydrochloride) [30]. Joseph et al. synthesized a positively charged amphiphilic pillar[5]arene that can inhibit biofilm formation in Gram-positive and Gram-negative bacteria [31]. Despite these advances, however, the effects of pillararenes on protein or enzyme bioactivity are incompletely understood [25,27].

Protein tyrosine kinases (PTKs) are important enzymes that catalyze the addition of phosphoryl group from adenosine triphosphate (ATP) to the tyrosine (Tyr) hydroxyl group, resulting in the phosphorylation of Tyr residues [32]. PTKs are divided into two groups: (1) membrane receptor PTKs (RTKs), such as epidermal growth factor receptor and insulin receptor, and (2) non-receptor-linked PTKs [33]. In general, RTKs undergo auto-phosphorylation via ligand-binding and trigger a cascade of phosphorylation events involving downstream kinases, which leads to the phosphorylation of target proteins in cells involved in cell proliferation, differentiation, and carcinogenesis [34]. Furthermore, many diseases (e.g., cancer, diabetes, and neurodegenerative disease) are connected to PTK activity [35,36]. Thus, targeting the active site of PTK via small molecules/drugs to inhibit Tyr phosphorylation remains a challenge [37,38]. Alfonso’s research group recently synthesized two artificial peptide cages that could inhibit Tyr phosphorylation by forming a supramolecular complex with Tyr residues in the polypeptide substrate [39]. On the basis of the highly inhibitory action of the PTK enzyme through substrate binding, we hypothesized that macrocyclic molecules may hinder the phosphorylation of Tyr residue due to their strong affinity to form complexes between the corresponding peptide substrate and supramolecule. However, to the best of our knowledge, the inhibition of Tyr phosphorylation by a typical macrocyclic molecule has not yet been reported.

Here, we first explored the ability of cationic pillar[6]arene (cPA6, Scheme 1) to inhibit Tyr phosphorylation and investigated the inclusion capacity of cPA6 with Tyr. As a new two-dimensional (2D) nanomaterial, black phosphorus nanosheets (BPNS) can enter cells and were used as a carrier of cPA6. To investigate whether cPA6 inhibits tyrosine phosphorylation in cells, cPA6 was conjugated with BPNS, and the phosphorylation level of protein in cells was evaluated in the presence of the BPNS@cPA6 composite (Figure 1).

## 2. Results and Discussion

### 2.1. Inhibition of Tyr Phosphorylation by Macrocyclic Molecules in Vitro

Some macrocyclic molecules, including β-cyclodextrin (β-CD), cucurbit[7]uril (CB7), sulfonated calix[8]arene (SCX8), water-soluble carboxylated pillar[5]arene (wPA5), cationic pillar[5]arene (cPA5), and cPA6, were tested to explore whether macrocyclic molecules are able to inhibit Tyr phosphorylation. The synthesis and characterization of pillararenes are reported in the Appendix A. Subsequently, the inhibition of Tyr phosphorylation was performed using poly(Glu,Tyr) sodium salt (pEY), a random polymer with a molecular weight (MW) of 20–50 kDa and a Glu:Tyr molar ratio of 4:1, as a model substrate (Appendix A). As shown in Figure 2B, although most of macrocyclic molecules could inhibit phosphorylation by PTK, the inhibitory activity of positively charged macrocyclic hosts (cPA5 and cPA6) were much higher than that of negatively charged/uncharged macrocyclic hosts (β-CD, CB7, wPA5, and SCX8), suggesting that the greater positive charge of the macrocyclic molecules may be beneficial to bind pEY. Therefore, the binding ability of the two cationic pillararenes to pEY was further tested by isothermal titration calorimetry (ITC).

Based on *K_D_* value, which reflects the average contact affinity of a macrocyclic molecule to each E4Y (one repeating unit of pEY) binding site, cPA6 showed a higher affinity toward pEY with a binding constant (*K_D_* = (2.64 ± 0.21) × 10^−6^ M, Figure 2C,D) than cPA5 (*K_D_* = (5.65 ± 0.82) × 10^−6^ M, Appendix A), indicating stronger interactions between the cPA6 and pEY. Therefore, cPA6 was used as an inhibitor of Tyr phosphorylation in the subsequent experiments. As shown in Figure 2, the inhibition rate of Tyr phosphorylation increased with increasing cPA6 concentrations from 0 to 100 μM, and the highest phosphorylation inhibition rate was 68.76% in the presence of 100 μM cPA6.

### 2.2. Host–Guest Interaction between cPA6 and pEY

The host–guest interactions between pEY and cPA6 were investigated by ultraviolet (UV), circular dichroism (CD), fluorescence, and nuclear magnetic resonance (NMR) technologies. The UV spectra showed that absorbance was decreased slightly and a minor redshift occurred when cPA6 was added to the aqueous solution of pEY (Figure 3A). The CD spectrum of pEY displayed a traditional random coil conformation at a negative peak of 198 nm (Figure 3B); however, this peak dramatically decreased in the presence of cPA6, indicating that the cPA6 remarkably affected the structure of the peptide. Moreover, the fluorescence titration of cPA6 with pEY generated different fluorescence emission spectra. With the addition of pEY, the fluorescence of cPA6 was obviously quenched (Appendix A), suggesting the interaction between cPA6 and pEY.

To understand the host–guest interaction, the cPA6/pEY complex was investigated at pD 7.2 by ^1^H NMR spectroscopy in Tris-DCl buffered D_2_O. As shown in Figure 3C, the resonance peaks related to the protons of pEY were markedly decreased in the presence of cPA6 compared with those of the free guest, possibly because the ring of Tyr residues entered the cavity of cPA6 to form an inclusion complex. These results further demonstrated that cPA6 could form an inclusion host–guest complex. To obtain further information of the host–guest complex, 2D nuclear overhauser effect spectroscopy (NOESY) spectroscopy of pEY and cPA6 was performed. Figure 4 shows the NOESY spectrum of an equimolar mixture of cPA6 with pEY in Tris-DCl buffer, which clearly displays NOE cross-peaks between the H_1_ and H_2_ of pEY and the aromatic proton moieties (H_a_) of cPA6. To gain detailed information on the binding modes of the host–guest complex, we performed molecular docking studies. The possible binding models of cPA6/pEY were simulated using the DOCK6.7 software package. Figure 5 shows the energy-minimized structure of the cPA6/pEY complex, in which the guest molecule is tightly wrapped with cPA6. The Tyr residues of pEY were clearly threaded into the cavity of the cPA6 molecule, and the host–guest complex was stabilized by the stacking and hydrophobic interactions between the benzene ring of Tyr and the circumjacent benzene rings of cPA6. In addition, the carboxylic groups of glutamic acid (Glu) residues bind tightly with the circumjacent trimethylamines located on the lower side of cPA6 through electrostatic interactions (Figure 5A,B). These molecular docking data are in agreement with the obtained ^1^H NMR result. 

### 2.3. Effect of cPA6 on Tyr Phosphorylation in Cells

As cPA6 can inhibit Tyr phosphorylation through the formation of the cPA6/pEY complex, it is of great interest to further explore whether cPA6 affects phosphorylation at the cellular level. BPNS is a new 2D nanomaterial that has been widely used in drug delivery, photothermal therapy, and other fields [40]. Herein, BPNS was used as a carrier of cPA6. We synthesized fluorescein isothiocyanate (FITC)-modified BPNS@cPA6 nanocomposite and investigated whether cPA6 can enter cells. The synthesized nanocomposite was characterized by thermogravimetric analysis (TGA), Raman spectroscopy, Fourier transform infrared (FTIR) spectroscopy, zeta potential measurement, transmission electron microscopy, energy dispersive X-ray spectrometry (EDX), and atomic force microscopy (AFM). The mass loss of BPNS@cPA6 (16.44%) was more than that of BPNS (3.26%) when the temperature reached 400 °C, indicating that 13.2% of cPA6 was successfully loaded on BPNS based on the difference of mass loss between BPNS and BPNS@cPA6 (Figure 6A). Accordingly, in the subsequent experiments, the concentration of BPNS@cPA6 was added in cell with the same concentration of cPA6. The slight shift of the Raman spectra of BPNS toward high wavenumber compared with bulk phosphorus (BP) is ascribed to the formation of ultrathin nanosheets [41]. The Raman spectra of BPNS and BPNS@cPA6 (Figure 6B) indicate that BPNS@cPA6 maintains the structure of BPNS. The FTIR spectrum of BPNS in Figure 6C shows stretching vibrations of P-O (1072 cm^−1^ and 1120 cm^−1^) and P=O (1620 cm^−1^). As for cPA6, the bands observed at 3015 cm^−1^, 2954 cm^−1^, and 2856 cm^−1^ are assigned to CH_2_ and CH_3_ stretching vibrations; the bands observed at 1626 cm^−1^, 1506 cm^−1^, 1482 cm^−1^, and 1404 cm^−1^ belong to phenyl plane bending vibrations; the bands observed at 1205 cm^−1^ is attributed to C-O-C stretching vibration; and the bands at 969 cm^−1^, 876 cm^−1^, and 782 cm^−1^ are assigned to phenyl out-of-plane bending vibrations (Figure 6C). These characteristic peaks appeared in the FTIR spectra of BPNS@cPA6, indicating the successful composition of the materials. The zeta potential of BPNS was positive, however, that of BPNS@cPA6 was negative, because of the introduced positive charge of cPA6 (Figure 6D,E). The EDX analysis results of BPNS and BPNS@cPA6 are illustrated in Figure 7C,D. As can be observed, only P was present in BPNS and C, O, and Br existed in BPNS@cPA6 composites aside from P. This indicates the successful preparation of the nanocomposite. The AFM images of the materials (Figure 7E,F) revealed that the thickness of BPNS is approximately 3 nm, and that of BPNS@cPA6 is approximately 4 nm. The increase in thickness is due to the introduction of cPA6 to BPNS. All these results supported the successful preparation of the BPNS@cPA6 nanocomposite. The phosphorylation level of proteins was investigated using Western blot to investigate whether cPA6 inhibits phosphorylation in cell. As shown in Figure 8A, BPNS or cPA6 alone did not inhibit Tyr phosphorylation, whereas BPNS@cPA6 remarkably inhibited the phosphorylation, because cPA6 from BPNS@cPA6 entered the cell. Although cPA6 could inhibit the phosphorylation of the Tyr of the polypeptide (Figure 2A), a lower inhibition of phosphorylation was observed in the presence of cPA6 alone, because less cPA6 entered cells when compared with BPNS@cPA6.

FITC-modified BPNS@cPA6 nanocomposite were added to the human hepatocellular carcinoma cell line HepG2 to detect the role of BPNS@cPA6 in cells. We can confirm that more BPNS@cPA6 nanocomposites entered the cells as the incubation time increased (Appendix A). Intriguingly, the addition of BPNS@cPA6 induced a concentration-dependent decrease in cell viability as evaluated by Counting Kit-8 (CCK-8) assay with a maximum inhibition rate of 64.50%. However, this reduced viability was not found in cells incubated with BPNS or cPA6 alone (Figure 8B). Apoptotic cells were stained by fluorescence assay using the Hoechst 33342 nucleic acid stain to determine whether cPA6 induces cell apoptosis. As shown in Figure 8C, the red dots stained by propidium iodide (PI) increased in the presence of BPNS@cPA6 compared with BPNS. This result indicated that BPNS@cPA6 induced apoptosis. Apoptosis detection was further confirmed using the Annexin V-FITC Cell Apoptosis Detection Kit. Similar results were obtained (Appendix A). All these results demonstrated that cPA6 induced apoptosis and decreased cell proliferation in HepG2 cells.

We further explored the effect of BPNS@cPA6 on the viability of normal and tumor cells using CCK-8 assay. As shown in Table 1, the half maximal inhibitory concentration (IC_50_) values of BPNS@cPA6 in normal cells were higher than those in tumor cells, indicating that BPNS@cPA6 has a stronger effect on tumor cells than on normal cells. PTK is more active in tumor cells than in normal cells [33], but phosphorylation was remarkably inhibited in the presence of BPNS@cPA6. This result may be caused by the formation of the tight host–guest interactions between cPA6 and Tyr, which inhibit Tyr phosphorylation. More negatively charged phosphatidylserine and phosphatidylinositol are found on the surface of tumor cells [42,43], which may promote the entrance of cPA6 into tumor cells and thus inhibit Tyr phosphorylation. BPNS@cPA6 may be used in tumor treatment because this composite can inhibit the proliferation of several tumor cells.

In this study, the inhibition of Tyr phosphorylation by cPA6 is based on supramolecular binding to the enzyme’s substrate instead of PTK activity. This mechanism is completely different from that of traditional small-molecule PTK inhibitors. Pre-existing or acquired mutations in the PTK domain induce drug resistance to small-molecule PTK inhibitors [44]. Thus, we believe that the drug resistance of small-molecule PTK inhibitors may be reduced when cPA6 is used as a phosphorylation inhibitor on the basis of the tight binding of cPA6 with the substrate/Tyr. Interestingly, as an easily synthesized and functionalized macromolecule [22,23,24,45,46] pillararenes can be functionalized to form supramolecular assemblies [47], and this feature may broaden the applications of this host in medical therapies, drug delivery, or other biological areas. Therefore, cPA6 may be an efficient anti-cancer drug that prevents ATP activity and inhibits Tyr phosphorylation. Herein, we presented a potential approach to modulate phosphorylation using a pillararene through substrate recognition. Although the inhibition of phosphorylation by cPA6 is nonspecific in cells, cPA6 may play a potential role in modulation of biological functions if the targeting design of materials is introduced in drug delivery.

## 3. Conclusions

In conclusion, we demonstrated the inclusion complexation behavior of macrocyclic hosts with pEY, which is an artificial substrate of PTK. Among the hosts assessed, cPA6 showed the strongest binding ability to pEY. Enzymatic Tyr phosphorylation was efficiently inhibited by cPA6, and the maximum phosphorylation inhibition rate (68.76%) was observed in the presence of 100 μM cPA6. Molecular docking results showed that the Tyr residues of pEY threaded into the cavity of cPA6 molecules and were stabilized through stacking and hydrophobic interactions between the benzene ring of Tyr and the circumjacent benzene rings of the host. In addition, the negatively charged carboxylic groups of Glu residues bound tightly with the circumjacent positively charged trimethylamines located on the lower side of cPA6 through electrostatic interactions. BPNS@cPA6 promoted cell apoptosis and inhibited cell proliferation by decreasing Tyr phosphorylation in HepG2 cells. BPNS@cPA6 also inhibited the viability of cells, especially tumor cells. These findings lay the groundwork for the use of pillararenes or their assemblies to modulate biochemical activity through substrate recognition

## 4. Materials and Methods

### 4.1. Materials, Reagents, and Equipment

SCX8 was obtained from Tokyo Chemical Industry Co., Ltd (Tokyo, Japan). β-CD, CB7, and pEY (MW = 20–50 kDa; concentration calculated on the basis of the MW of the repeating unit: 786) were purchased from Sigma-Aldrich Co., Ltd. (St. Louis, MO, USA). Phosphate-buffered saline (PBS, pH 7.2) was purchased from Shanghai BasalMedia Technologies Co., Ltd. (Shanghai, China). Black Universal Tyrosine Kinase Assay Kit was purchased from Takara Bio Inc. (Shiga, Japan). BP crystal was purchased from Kunming Black Phosphorus Technology Service Co., Ltd. (Kunming, China). FITC-PEG-NH_2_ was purchased from Aladdin Chemical Reagent Lo., Ltd. (Shanghai, China). Hoechst 33342, PI and Annexin V-FITC Cell Apoptosis Detection Kit were purchased from Beyotime Biotechnology Co., Ltd. (Shanghai, China). CCK-8 was purchased from DOJINDO Chemical Technology Co., Ltd. (Shanghai, China). Dulbecco’s modified Eagle medium (DMEM), fetal bovine serum (FBS), streptomycin, penicillin radioimmunoprecipitation assay (RIPA) buffer, EDTA-free protease inhibitor cocktail tablets and bicinchoninic acid (BCA) protein assay kit were purchased from Thermo Fisher Scientific (Waltham, MA, USA). Horseradish peroxidase (HRP-)-conjugated monoclonal anti-phospho-tyrosine antibody was purchased from R&D Systems Inc. (Minneapolis, MN, USA). Anti-actin monoclonal antibody was obtained from Affinity Biosciences Inc., (Cincinnati, OH, USA). HRP-conjugated secondary antibody was purchased from Santa Cruz Biotechnology Inc., (Delaware Ave Santa Cruz, CA, USA). All other reagents and solvents were commercially available and used as supplied without further purification. All aqueous solutions were prepared with deionized water (18 MΩ cm).

UV spectra were analyzed in a UNICO 2100 UV–visible spectrometer (Shanghai, China). CD spectra were measured at 190–400 nm with a Chirascan spectrometer (Applied Phomiddlehysics Ltd., Surrey, UK), and the digitized data were transferred to a microcomputer and processed. Samples were dissolved in PBS (pH 7.2) at a concentration of 20 μM. ITC experiments were performed using a MicroCal PEAQ-ITC colorimeter (MicroCal, Northampton, MA, USA) to investigate the thermodynamic and kinetic characteristics of the interactions between pEY and pillararenes. NMR spectra were recorded with a Brucker AVANCE DRX 400, a Bruker AVANCE III HD 500 or a Bruker AVANCE III HD 600 spectrophotometer (Rheinstetten, Germany). The absorbance of a 96-well microplate was read on a BioTek Epoch (Winooski, VT, USA). A Q50 TGA instrument (New Castle, DE, USA) was used for TGA. Raman spectra were obtained on a 400F Perkin-Elmer Raman spectrometer (Shelton, WA, USA) with a 514.5 nm wavelength incident laser light. A Thermo Fisher Scientific Nicolet IS10 FTIR (Waltham, MA, USA) or Nicolet Impact 410 FTIR spectrophotometer was used for the FTIR spectrometry. A Malvern Zetasizer Nano series was used for zeta potential measurements. The morphologies of the prepared samples were characterized by a Bruker Dimension Edge atomic force microscope (Santa Barbara, CA, USA) and a JEM 2100 transmission electron microscope (Tokyo, Japan) equipped with an energy dispersive X-ray spectrometer. Polyvinylidene fluoride (PVDF) membranes were sourced from Millipore (Billerica, MA, USA). All cells were visualized by IN Cell Analyzer 2200^TM^ (GE Healthcare Bio-Sciences, Little Chalfont, UK). Immunoreactive bands were visualized by Amersham Imager 6000 (GE, Tokyo, Japan).

### 4.2. Effect of Macrocyclic Supramolecules on Tyr Phosphorylation

The inhibition rate of Tyr phosphorylation was determined with a Universal Tyrosine Kinase Assay Kit. Each supramolecule of cPA6 (100 μM), cPA5 (400 μM), SCX8 (400 μM), wPA5 (400 μM), CB7 (400 μM), or β-CD (400 μM) was dissolved in PBS (pH 7.2) and added to the wells containing pEY substrate at 25 °C for 1 h before the addition of PTK and ATP. Then, the free supramolecule was removed by washing with PBS for three times. The remaining experimental steps followed the manufacturer’s instructions. Each experiment was performed in triplicate.

### 4.3. Molecular Docking

The X-ray structure of cPA6 analog (COD code: 41256564) was downloaded from the Crystallography Open Database (http://www.crystallography.net/), and UCSF Chimera was used to construct the 3D structure of cPA6. Hydrogen atoms were added and AM1-BCC charges assigned by using the Dock Prep module with subsequent energy minimization. The initial 3D structure of the peptide repeated unit pEY was also constructed in Chimera and then energy minimization was performed at an AMBER ff14SB force field to obtain the optimized structure. The DMS tool in Chimera was employed to generate the molecular surface of cPA6 using a probe atom with a 1.4 Å radius. The box center was set as x = 5.146 Å, y = 17.124 Å, and z = 10.322 Å, and lengths in three directions were as x = 28.333 Å, y = 30.209 Å, and z = 26.263 Å. The sphgen module was used to generate spheres that fill the surface because the whole cPA6 structure was treated as the binding site. The Grid module was then used to produce Grid files for energy evaluation. Semi-flexible dockings were conducted in DOCK program, where 10,000 different orientations were generated. Van der Waals and electrostatic interactions were obtained between the host and guest. Afterward, clustering analysis was performed (RMSD threshold was set to 2.0 Å) to obtain the best scored poses. Finally, PyMOL was used for the visualization of the docking results.

### 4.4. Synthesis of BPNS

Bulk BP (100 mg) dispersed in oxygen-free water (50 mL) was sonicated in an ice bath for 10 h. Then, the suspension of BPNS was obtained by centrifugation at 2000 rpm for 10 min to dislodge the unexfoliated bulk BP. The whole process was conducted in a light-free environment.

### 4.5. Synthesis of BPNS @cPA6

BPNS (20 mg) and cPA6 (20 mg) were dispersed in oxygen-free water (10 mL) and sonicated in an ice bath for 20 min. After stirring for 4 h, the mixture was centrifuged thrice to remove excess cPA6. The whole process was carried out in a light-free environment.

### 4.6. Synthesis of BPNS-PEG-FITC and BPNS @cPA6-PEG-FITC

FITC-PEG-NH_2_ (2 mg) and BPNS (0.2 mg) were dispersed in oxygen-free water (20 mL). The mixture was sonicated in an ice bath for 20 min, stirred at 4 °C for 4 h, and centrifuged thrice to remove excess FITC-PEG-NH_2_ molecules. BPNS@cPA6-PEG-FITC was prepared using the same procedure as BPNS@cPA6 instead of BPNS. The whole process was carried out in a light-free environment.

### 4.7. Cell Culture

HepG2 cells were cultured in DMEM containing 10% FBS and 1% penicillin/streptomycin. Cells were cultured at 37 °C and 5% CO_2_.

### 4.8. Uptake of BPNS@cPA6 in Cells

HepG2 cells at a density of 5 × 10^4^ cells per well were seeded in six-well culture plates for 24 h. Then, the cells were incubated with BPNS-PEG-FITC and BPNS@cPA6-PEG-FITC (0.02 mg·mL^−1^) at 37 °C under 5% CO_2_ for different durations (1, 3, and 12 h). Then, the cell nuclei were labeled with Hoechst 33342 solution (100×, 0.02 mL·well^−1^) for 10 min. All cells were visualized by IN Cell Analyzer 2200.

### 4.9. Western Blot

Cells were dissolved by Radio-Immunoprecipitation Assay (RIPA) buffer with ethylenediaminetetraacetic acid (EDTA)-free protease inhibitor cocktail tablets. The concentration of total protein was determined using the BCA protein assay kit. The cell lysate was subjected to sodium dodecyl sulfate-polyacrylamide gel electrophoresis and transferred onto PVDF membranes. The membranes were then incubated with the following primary antibodies: HRP-conjugated monoclonal anti-phospho-tyrosine antibody (1:5000), and anti-actin monoclonal antibody (1:3000). Subsequently, the membranes were incubated with HRP-conjugated secondary antibody (1:5000) and the immunoreactive bands were visualized by SuperSignal West Pico Chemiluminescent Substrate using Amersham Imager 6000.

### 4.10. Cell Viability

HepG2 cells were seeded in a 96-well plate (1 × 10^4^ cells·well^−1^) in DMEM at 37 °C under 5% CO_2_ for 24 h. Then, the cells were incubated with different concentrations of BPNS, BPNS@cPA6 (0, 0.033, 0.083, 0.167, 0.25, and 0.333 mg·mL^−1^) and cPA6 (0, 0.004, 0.011, 0.022, 0.033, and 0.0440 mg·mL^−1^) for 24 h. A standard CCK-8 assay was used to determine cell viability. Hoechst 33342 (100 ×, 0.002 mL·well^−1^) and PI (100×, 0.002 mL·well^−1^) solutions were used to stain apoptotic cells, which were also visualized by IN Cell Analyzer 2200. We also used a standard Annexin V-FITC Cell Apoptosis Detection Kit assay to determine apoptosis and visualized by IN Cell Analyzer 2200. HepG2 cells were incubated with PBS, BPNS (0.1 mg·mL^−1^ or BPNS@cPA6 (0.1 mg·mL^−1^) in DMEM for 24 h.

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
