# Peer review of "Cationic Pillar[6]arene Induces Cell Apoptosis by Inhibiting Protein Tyrosine Phosphorylation Via Host–Guest Recognition"

_ijms, 2020, doi:10.3390/ijms21144979_

Round 1

Reviewer 1 Report

The Manuscript entitled “Cationic pillar[6]arene induces cell apoptosis by inhibiting protein tyrosine phosphorylation via host-guest recognition” reported the use of a cationic pillar[6]arene to inhibit tyrosine moieties phosphorylation via molecular recognition.

The manuscript is well written and the scientific part soundly discussed.

Nevertheless, some comments arose.

  1. In the keywords the term “pillar[6]arene is missing
  2. In the introduction, the contribute of cavitands to the field is missing. Other contribute to the field are missing. In particular, the following papers are worth citing:
  3. a) R. Pinalli, A. Pedrini and Dalcanale E., Chem Soc. Rev. 2018, 47, 7006-7026.
  4. b) N. Bontempi, E. Biavardi, D. Bordiga, G. Candiani, I. Alessandri, P. Bergese and E. Dalcanale, Nanoscale, 2017, 9, 8639–8646
  5. c) Y. Liu, L. Perez, M. Mettry, C. J. Easley, R. J. Hooley and W. Zhong, Am. Chem. Soc., 2016, 138, 10746–10749
  6. d) Y. Liu, L. Perez, A. D. Gill, M. Mettry, L. Li, Y. Wang, R. J. Hooley and W. Zhong, Am. Chem. Soc., 2017, 139, 10964–10967
  7. e) S. A. Minaker, K. D. Daze, M. C. F. Ma and F. Hof, Am. Chem. Soc., 2012, 134, 11674–11680.
  8. f) R. Zadmard and T. Schrader, Am. Chem. Soc., 2005, 127, 904–915.
  9. g) F. Guagnini, P. M. Antonik, M. L. Rennie, P. O’Byrne, A. R. Khan, R. Pinalli, E. Dalcanale, and P. B. Crowley, Chem. Int. Ed., 2018, 57, 7126 –7130
  10. In Figure 2B as well as in the text, the concentrations of the macrocycles are not explained.
  11. The Reviewer did not agree with the claim, based on the KD values, that cPA6 has higher affinity towards pEY respect to cPA5. In fact, the KD order of magnitude 10-6 is the same for both the macrocycles, and a difference of 3·10-6 without the error value is not significant. Please provide the error, the DH and DS values obtained by ITC and eventually correct the sentence
  12. Page 5 line 7 and 13. It is not clear to the Reviewer if the NMR is done in D2O, as written in the text, or Tris-DCl buffer, as written in the caption of Figure 3C.
  13. The sentence page 6 line 8 is not clear. What do the Authors mean with: “The phenolic group of the Tyr residue of pEY may also enter the ring of cPA6 and form a supramolecular complex.” Before they claimed that “The Tyr residues of pEY were clearly threaded into the cavity of the cPA6 molecule,……”. Do they mean that also the second Tyr can be complexed by the macrocycle without any steric hindrance? Please explain better the sentence or remove it.
  14. The NOESY spectrum reported in Figure 4 page 6 is absolutely unclear. In the opinion of the Referee the f1 and f2 axes are shifted since the diagonal is not centered on the peaks. The cross peak between H2 of pEY and Ha of cPA6 claimed by the Authors is not present on both the side of the diagonal and the one highlighted by the two red lines has the same intensity of the noise. Correct also the caption: D2O or Tris buffer? Correct as well 25°C.
  15. Page 7 line 8: the ref 33 is not correct. The article does not mention BPNS. The same holds for ref 34. The Referee suggests a careful revision of all the references cited.
  16. In Figure 6A the TGA of cPA6 is missing. please provide it to complete the thermal characterization. Explain if it is under N2 or air. The caption of Figure 6 is completely wrong. it is the same of Figure 7. Please change it
  17. Page 10, line 5-6 the sentence “This result indicated that the co-localization of apoptotic cells and BPNS@cPA6.” has no sense. please provide to correct sentence.
  18. Page 10 line 20: what is the number 26 superscript on the word cells? A ref? Please correct
  19. The speculation reported by the Author at page 11 lines11-25 has no sense. Some part of the sentence, where references are reported, can be translated to the introduction, but the rest of the sentence must be delated. Moreover, line 19 the Author claimed “We speculate that cPA6 is capable of potential binding with Tyr residues and may inhibit intracellular phosphorylation by PTK.” I thought that the manuscript demonstrated it. rephrase it and delete all the speculations.
  20. 12. page 2 line 20 the ref 32 is wrong.

Grammar errors:

  1. 1. Abstract, line 18. Wrong spacing between the words phosphorylation and through
  2. Page 2 line 36 the sentence in the bracket (Scheme S1-S12, Figure S1-S10) has no sense there. The suggestion is to write for example: The synthesis and characterization of pillarenes are reported in etc……
  3. Page 11 line 19: after the word weakened there is a comma instead of a dot.

Author Response

Responses to the Comments made by the reviewers 1

The Manuscript entitled “Cationic pillar[6]arene induces cell apoptosis by inhibiting protein tyrosine phosphorylation via host-guest recognition” reported the use of a cationic pillar[6]arene to inhibit tyrosine moieties phosphorylation via molecular recognition. The manuscript is well written and the scientific part soundly discussed.
Nevertheless, some comments arose.
1. In the keywords the term “pillar[6]arene is missing.
Response 1: Thanks. We changed keyword “macrocyclic molecules” to “pillar[6]arene” according to the reviewer’s advice.

2. In the introduction, the contribute of cavitands to the field is missing. Other contribute to the field are missing. In particular, the following papers are worth citing:
3. a) R. Pinalli, A. Pedrini and Dalcanale E., Chem Soc. Rev. 2018, 47, 7006-7026.
4. b) N. Bontempi, E. Biavardi, D. Bordiga, G. Candiani, I. Alessandri, P. Bergese and E. Dalcanale, Nanoscale, 2017, 9, 8639–8646
5. c) Y. Liu, L. Perez, M. Mettry, C. J. Easley, R. J. Hooley and W. Zhong, Am. Chem. Soc., 2016, 138, 10746–10749
6. d) Y. Liu, L. Perez, A. D. Gill, M. Mettry, L. Li, Y. Wang, R. J. Hooley and W. Zhong, Am. Chem. Soc., 2017, 139, 10964–10967
7. e) S. A. Minaker, K. D. Daze, M. C. F. Ma and F. Hof, Am. Chem. Soc., 2012, 134, 11674–11680.
8. f) R. Zadmard and T. Schrader, Am. Chem. Soc., 2005, 127, 904–915.
9. g) F. Guagnini, P. M. Antonik, M. L. Rennie, P. O’Byrne, A. R. Khan, R. Pinalli, E. Dalcanale, and P. B. Crowley, Chem. Int. Ed., 2018, 57, 7126 – 7130
Response 2: As the comments 2-9, we added these citations in the revised manuscript according to the reviewer’s advice.

10. In Figure 2B as well as in the text, the concentrations of the macrocycles are
not explained.
Response 3: Thanks.To help the reader understand the content easily, the
concentrations of macrocycles were moved to the legend of Fig. 2.

11. The Reviewer did not agree with the claim, based on the KD values, that cPA6 has higher affinity towards pEY respect to cPA5. In fact, the KD order of
magnitude 10-6 is the same for both the macrocycles, and a difference of 3×10-6 without the error value is not significant. Please provide the error, the H and S values obtained by ITC and eventually correct the sentence.
Response 4: Thanks. We added SD values of KD, H, and S values in the Fig. 2 according to the reviewer’s advice.

12. Page 5 line 7 and 13. It is not clear to the Reviewer if the NMR is done in D2O, as written in the text, or Tris-DCl buffer, as written in the caption of Figure
3C.
Response 5: Thank you for your comment. Herein, the NMR was done in Tris-DCl buffer prepared with D2O. So we revised the manuscript to make it clear.

13. The sentence page 6 line 8 is not clear. What do the Authors mean with: “The phenolic group of the Tyr residue of pEY may also enter the ring of cPA6 and form a supramolecular complex.” Before they claimed that “The Tyr residues
of pEY were clearly threaded into the cavity of the cPA6 molecule,……”. Do they mean that also the second Tyr can be complexed by the macrocycle without any steric hindrance? Please explain better the sentence or remove it.
Response 6: Thank you for your comment. We deleted this sentence according to the referee’s advice.

14. The NOESY spectrum reported in Figure 4 page 6 is absolutely unclear. In the opinion of the Referee the f1 and f2 axes are shifted since the diagonal is not centered on the peaks. The cross peak between H2 of pEY and Ha of cPA6 claimed by the Authors is not present on both the side of the diagonal and the one highlighted by the two red lines has the same intensity of the noise. Correct also the caption: D2O or Tris buffer? Correct as well 25°C.
Response 7: Thank you for your comment. We redrew the Fig. and revised the
manuscript according to the referee’s advice.

15. Page 7 line 8: the ref 33 is not correct. The article does not mention BPNS. The same holds for ref 34. The Referee suggests a careful revision of all the references cited.
Response 8: Thank you for your careful comments. We adjusted the citations and revised the manuscript according to the referee’s advice.

16. In Figure 6A the TGA of cPA6 is missing. please provide it to complete the thermal characterization. Explain if it is under N2 or air. The caption of Figure 6 is completely wrong. it is the same of Figure 7. Please change it
Response 9: We made mistakes. It was corrected.

17. Page 10, line 5-6 the sentence “This result indicated that the co-localization of apoptotic cells and BPNS@cPA6.” has no sense. please provide to correct sentence.
Response 10: Thanks. We revised the sentence according to the referee’s advice.

18. Page 10 line 20: what is the number 26 superscript on the word cells? A ref? Please correct
Response 11: Thanks. This is a citation number. We checked and confirmed this
citation number.

The speculation reported by the Author at page 11 lines11-25 has no sense. Some part of the sentence, where references are reported, can be translated to the introduction, but the rest of the sentence must be delated. Moreover, line 19 the Author claimed “We speculate that cPA6 is capable of potential binding with Tyr residues and may inhibit intracellular phosphorylation by PTK.” I thought that the manuscript demonstrated it. rephrase it and delete all the speculations.
Response 12: Thank you for your constructive suggestion. We deleted some sentence and rewrote the discussion according to the referee’s advice.

19. 12. page 2 line 20 the ref 32 is wrong.
Response 13: All the references were checked and adjusted.

Grammar errors:
1. 1. Abstract, line 18. Wrong spacing between the words phosphorylation and
through
Response 14: Thanks. This was corrected.

2. Page 2 line 36 the sentence in the bracket (Scheme S1-S12, Figure S1-S10)
has no sense there. The suggestion is to write for example: The synthesis and
characterization of pillarenes are reported in etc……
Response 15: This sentence was corrected.

3. Page 11 line 19: after the word weakened there is a comma instead of a dot.
Response 16: This sentence was deleted.

Reviewer 2 Report

The manuscript “Cationic pillar[6]arene induces cell apoptosis by inhibiting protein tyrosine phosphorylation via host-guest recognition” by Can-Peng Li, Yuxun Lu, Chengting Zi, Yuting Zhao, Hui Zhao, and Ya-Ping Zhang presents a pillararene based nanocomposite BPNS@cPA6 that can induce apoptosis and inhibit cell proliferation by reducing the level of Tyrosine phosphorylation through host-guest chemistry.

This comprehensive study was tested with different macrocycles from a variety of receptor families (cyclodextrin, cucurbituril, calixarene, and pillararene) with different properties. All the other macrocycles were either neutral or anionic. For the three pillararenes, one was anionic and the other cPA5 of a smaller internal cavity. The pEY is anionic in nature. Clearly, the key attractive forces were electrostatic and also size complementarity between the Tyr and the cPA6.

Multiple techniques were used to analyze the different processes and support the conclusion from the study. This is very good work with some very promising results.

However, I will defer acceptance since I think some revisions are needed.

1) The references should be revised completely. The citations in the text did not match the correct references in numerous cases. For example, check reference 22-24 and compare to the citations.

2) Figure 2B: I believe the concentrations for all the other receptors are also 100 uM? Indicate that in the figure legend.

3) I believe the ITC fittings in Figure 2D is from Figure 2C. Many points at the beginning of the fitting do not match figure 2C. Please check again.

4) Page 4, first paragraph: What is E4Y?

5) Figure 6A legend is wrong. Should be TGA description.

6) Figure 6B. BP bulk not BP Bluk??

7) Page 10, first paragraph. “However, this reduced viability was not found in cells incubated with BPNS or cPA6 alone (Figure 8B).” This throws me off. Why don’t we see any reduced viability with the pure cPA6? This host should still be able to bind the Tyr on the polypeptide and therefore block it from phosphorylation. The docking study clearly suggests that as well. A clear and detailed explanation should be provided.

Author Response

Responses to the Comments made by the reviewers2

The manuscript “Cationic pillar[6]arene induces cell apoptosis by inhibiting protein tyrosine phosphorylation via host-guest recognition” by Can-Peng Li, Yuxun Lu, Chengting Zi, Yuting Zhao, Hui Zhao, and Ya-Ping Zhang presents a pillararene based nanocomposite BPNS@cPA6 that can induce apoptosis and inhibit cell proliferation by reducing the level of Tyrosine phosphorylation through host-guest chemistry.

This comprehensive study was tested with different macrocycles from a variety of receptor families (cyclodextrin, cucurbituril, calixarene, and pillararene) with different properties. All the other macrocycles were either neutral or anionic. For the three pillararenes, one was anionic and the other cPA5 of a smaller internal cavity. The pEY is anionic in nature. Clearly, the key attractive forces were electrostatic and also size complementarity between the Tyr and the cPA6.
Multiple techniques were used to analyze the different processes and support the conclusion from the study. This is very good work with some very promising results.

However, I will defer acceptance since I think some revisions are needed.
1) The references should be revised completely. The citations in the text did not match the correct references in numerous cases. For example, check reference 22-24 and compare to the citations.

Response 17: Thank you for your comment. We checked all the citations and uniformed the styles according to the reviewer’s advice.

2) Figure 2B: I believe the concentrations for all the other receptors are also 100 uM? Indicate that in the figure legend.
Response 18: Thank. The concentration of macrocyclic supramolecules were given in section 4.2. We removed the concentration of macrocyclic supromolecules from Experimental section to Fig. 2 legend according to the reviewer’s advice.

3) I believe the ITC fittings in Figure 2D is from Figure 2C. Many points at the beginning of the fitting do not match figure 2C. Please check again.
Response 19: Yes, ITC fittings in Figure 2D is from Figure 2C. In Fig. 2C, the first point is a blank, and the numbers of injection (pEY solution) into cPA5 solution are 18, and therefore 18 H values were obtained. We checked the content and confirm it is OK.

4) Page 4, first paragraph: What is E4Y?
Response 20: Thanks. E4Y is a repeating unit of pEY, a substrate for PTK. We rewrote this sentence to make it clear.

5) Figure 6A legend is wrong. Should be TGA description.
Response 21: We made a mistake. So we revised the legend according to the reviewer’s advice.

6) Figure 6B. BP bulk not BP Bluk??
Response 22: Thank you for your comment. We made a mistake. This was corrected.

7) Page 10, first paragraph. “However, this reduced viability was not found in cells incubated with BPNS or cPA6 alone (Figure 8B).” This throws me off. Why don’t we see any reduced viability with the pure cPA6? This host should still be able to bind the Tyr on the polypeptide and therefore block it from phosphorylation. The docking study clearly suggests that as well. A clear and detailed explanation should be provided.
Response 23: Thanks. In this work, indeed cPA6 can bind to Tyr on the peptide, however cPA6 itself may not enter cells and cannot inhibit intracellular Tyr phosphorylation. Therefore, as a design strategy, cPA6 was first conjugated with BPNS and a cPA6/BPNS nanocomposite was prepared. Then the cPA6/BPNS was added to cells. The cPA6/BPNS can enter cells due to the endocytosis effect, and played the inhibition of Tyr phosphorylation by cPA6 molecule. This was explained clearly in the text.
Response 29: We made a mistake. So the word “BPNS@cPA6” was used throughout the text.

Results and discussion:
• In figure 2, the hexagonal prism illustrating the cPA6 molecule looks rather like a crystal. This could be improved by using a truncated cone, which is probably a more realistic representation of the pillarene molecule.
Response 30: Here, hexagonal prism was used for cPA6 molecule in Fig. 1, since cPA6 is a highly symmetric molecule (Scheme 1).
• Page 3, line 6: "Try" -> "Tyr".
Response 31: We made a mistake. This was corrected.
• Page 5, line 4: "1H NMR" -> "1H NMR".
Response 32: Corrected.
• Page 6, line 11: "D2O, 25 oC" -> "D2O, 25 °C".
Response 33: Corrected.
• Page 8: The caption to Figure 6 appears to be incorrect and in (c), the abscissa should be "Wavenumber / cm-1" rather than "Wavelength / nm", I reckon.
Response 34: We made a mistake. This has been corrected.
• Figure 8 (c): "Hochest" -> "Hoechst".
Response 35: Corrected.
• Page 11, line 18: add a comma before "because".
Response 36: These sentences were deleted.
Page 11, line 19: replace the comma after "weakened" by a full stop.
Response 37: Corrected.
• Page 11, line 22: "using pillarene" -> "using a pillarene".
Response 38: Revised.

Experimental:
• Page 14, line 9: it should be mentioned here what kind of supporting information is available.
Response 39: Thanks. The supporting information has been uploaded in Submission system according to the request of the Journal. We think if the manuscript is accepted, a final web site will be given.

Reviewer 3 Report

Can-Peng Li et al. studied a cationic pillar[6]arene as inhibitor of tyrosine phosphorylation. I recommend publication of this manuscript in International Journal of Molecular Sciences after the following minor corrections:

Abstract: It is actually a matter of style, but an abstract should always be written in present tense and one should not decribe what one did, e.g. not "We synthesized..." but what the key points are.

Introduction:

  • Refs. 1-5 are 15 to 20 years old. I wonder whether there is some more recent literature that could be cited here.
  • Page 1, line 34: "Emerging studies" should probably be "Recent studies".
  • Showing a chemical diagram of a pillar[6]arene in the introduction would be helpful.
  • Page 2, line 5: the Barbera reference is actually ref. 21 in the list, not ref. 23. So I recommend a careful check of the numbering of all references in the entire manuscript.
  • Page 2, line 9: it shoud actually read "... that catalyze the transfer of a phosphate group from...". A phosphoryl group is something different.
  • Page 2, line 30: here it is "cPA6@BPNS" but otherwise it is "BPNS@cPA6". Since cPA6 was loaded onto BPNS, the first version seems natural to me. In some places, there is a space before or after the @. In any case, it should be consistent throughout.

Results and discussion:

  • In figure 2, the hegagonal prism illustrating the cPA6 molecule looks rather like a crystal. This could be improved by using a truncated cone, which is probably a more realistic representation of the pillarene molecule.
  • Page 3, line 6: "Try" -> "Tyr".
  • Page 5, line 4: "1H NMR" -> "1H NMR".
  • Page 6, line 11: "D2O, 25 oC" -> "D2O, 25 °C".
  • Page 8: The caption to Figure 6 appears to be incorrect and in (c), the abscissa should be "Wavenumber / cm-1" rather than "Wavelength / nm", I reckon.
  • Figure 8 (c): "Hochest" -> "Hoechst".
  • Page 11, line 18: add a comma before "because".
  • Page 11, line 19: replace the comma after "weakened" by a full stop.
  • Page 11, line 22: "using pillarene" -> "using a pillarene".

Experimental:

  • Page 14, line 9: it should be mentioned here what kind of supporting information is available.

Author Response

Responses to the Comments made by the reviewers3

Can-Peng Li et al. studied a cationic pillar[6]arene as inhibitor of tyrosine phosphorylation. I recommend publication of this manuscript in International Journal of Molecular Sciences after the following minor corrections:

Abstract: It is actually a matter of style, but an abstract should always be written in present tense and one should not decribe what one did, e.g. not "We synthesized..." but what the key points are.
Introduction:
• Refs. 1-5 are 15 to 20 years old. I wonder whether there is some more recent literature that could be cited here.
Response 24: Thank you for your comment. We added some recent citations in the revised manuscript according to the referee’s advice.

• Page 1, line 34: "Emerging studies" should probably be "Recent studies".
Response 25: Thanks. This was changed as requested.

• Showing a chemical diagram of a pillar[6]arene in the introduction would be helpful.
Response 26: Thanks. We added chemical diagram of a pillar[6]arene in the text (Scheme 1).

• Page 2, line 5: the Barbera reference is actually ref. 21 in the list, not ref. 23. So I recommend a careful check of the numbering of all references in the entire manuscript.
Response 27: Thank you for your suggestion. We checked and adjusted the citations in the text entirely.

• Page 2, line 9: it shoud actually read "... that catalyze the transfer of a phosphate group from...". A phosphoryl group is something different.
Response 28: We revised this sentence as to “catalyze the addition of phosphate to tyrosine residues”.

• Page 2, line 30: here it is "cPA6@BPNS" but otherwise it is "BPNS@cPA6". Since cPA6 was loaded onto BPNS, the first version seems natural to me. In some places, there is a space before or after the @. In any case, it should be consistent throughout.
Response 29: We made a mistake. So the word “BPNS@cPA6” was used throughout the text.

Results and discussion:
• In figure 2, the hexagonal prism illustrating the cPA6 molecule looks rather like a crystal. This could be improved by using a truncated cone, which is probably a more realistic representation of the pillarene molecule.
Response 30: Here, hexagonal prism was used for cPA6 molecule in Fig. 1, since cPA6 is a highly symmetric molecule (Scheme 1).

• Page 3, line 6: "Try" -> "Tyr".
Response 31: We made a mistake. This was corrected.

• Page 5, line 4: "1H NMR" -> "1H NMR".
Response 32: Corrected.

• Page 6, line 11: "D2O, 25 oC" -> "D2O, 25 °C".
Response 33: Corrected.

• Page 8: The caption to Figure 6 appears to be incorrect and in (c), the abscissa should be "Wavenumber / cm-1" rather than "Wavelength / nm", I reckon.
Response 34: We made a mistake. This has been corrected.

• Figure 8 (c): "Hochest" -> "Hoechst".
Response 35: Corrected.

• Page 11, line 18: add a comma before "because".
Response 36: These sentences were deleted.

Page 11, line 19: replace the comma after "weakened" by a full stop.
Response 37: Corrected.

• Page 11, line 22: "using pillarene" -> "using a pillarene".
Response 38: Revised.

Experimental:
• Page 14, line 9: it should be mentioned here what kind of supporting information is available.
Response 39: Thanks. The supporting information has been uploaded in Submission system according to the request of the Journal. We think if the manuscript is accepted, a final web site will be given.

Round 2

Reviewer 2 Report

The authors did a good job addressing all my concerns. I, therefore, recommend acceptance.